# A new target: AlkBH2 promotes bladder cancer by upregulation of inflammation

Zhangjie Yang[1]*, Jinhu Ma[1], Ziyang Qiang[1,2], Wenhao Xie[1], Liang Jiao[1], Guojun Chen[1,2]*

**1** Clinic Medicine College of Qinghai University, Xining, China, **2** Department of Urology, Affiliated Hospital of Qinghai University, Xining, China

* chenguojun68@126.com (GC); aa67316693@163.com (ZY)

## Abstract

A close relationship exists between inflammation and cancer. Recent studies have highlighted inflammation as a significant contributor to the progression of bladder cancer. However, the role of alkyladenine DNA glycosylase homolog 2 (AlkBH2), an enzyme involved in DNA repair and a member of the AlkB family, in the context of bladder cancer inflammation remains largely unexplored. Our findings demonstrate that AlkBH2 promotes the proliferation, colony formation, migration, and invasion of bladder cancer cells. Mechanistically, AlkBH2 activates the nuclear factor-kappa B (NF-κB) signaling pathway, which in turn drives the progression of bladder cancer. These results suggest that AlkBH2 plays a critical oncogenic role in bladder cancer by modulating inflammation through the activation of the NF-κB pathway. These findings highlight the potential of AlkBH2 as a therapeutic target for bladder cancer treatment.

## Introduction

According to the World Health Organization (WHO) report from 2024, bladder cancer ranks as the 9th most common cancer globally, accounting for over 220,000 annual deaths [1]. As a malignancy of the bladder mucosa, bladder cancer is the most prevalent urinary system tumor and is among the top ten most common cancers in China [2]. It is characterized by a higher incidence in men in developed regions, and the number of new cases is increasing annually worldwide [3]. Once muscular invasion occurs, bladder cancer tends to metastasize readily, leading to a poor prognosis [4]. These findings underscore the persistent global challenge posed by bladder cancer and highlight the urgent need for additional research and preventive strategies. Consequently, elucidating novel mechanisms and identifying therapeutic targets within the complex regulatory network of bladder cancer remains of paramount importance.

Chronic inflammation is a well-established enabler of carcinogenesis across multiple cancer types, including bladder cancer [5]. The relationship between systemic inflammatory factors and bladder cancer risk, however, is not fully

**Data availability statement:** All relevant data are within the paper and its Supporting information files.

**Funding:** Qinghai Key Construction Project of Specialized Departments of Qinghai University Affiliated Hospital (Qinghai Weijianwei [2023]133).

**Competing interests:** The authors have declared that no competing interests exist.

delineated, with observational studies presenting conflicting results. Some report positive correlations between elevated pro-inflammatory cytokines (e.g., IL-1, TNF-α) and increased risk, while others find no significant association, indicating the need for more nuanced investigation [6]. Clinically, the pre-surgical inflammatory environment in bladder cancer patients is closely associated with post-surgical prognosis [7]. This clinical observation is rooted in the fundamental biology of bladder cancer, which is recognized as an inflammation-linked disease [8]. The tumor microenvironment consists of various stromal tissues, inflammatory cells, infiltrating immune cells, and soluble mediators where inflammation and cancer progression are intimately linked. Inflammatory signals can drive the initial onco-genic transformation and subsequent tumor growth, while the tumor itself actively fosters a pro-inflammatory milieu that enhances its survival, invasiveness, and metastatic potential [9].

The AlkB family of dioxygenases, including AlkBH2, plays a crucial role in DNA repair [10]. Early studies in *Escherichia coli* demonstrated that AlkB mutants exhibited hypersensitivity to alkylating agents [11]. Humans possess nine AlkB family isoenzymes (AlkBH1–8 and FTO) [12]. AlkBH2 is a key dioxygenase responsible for repairing alkylation damage in genomic DNA, particularly in ribosomal DNA genes and double-stranded DNA [13]. Its clinical significance in human cancers is increasingly recognized [14]. While AlkBH2 has been shown to modulate chemotherapy sensitivity in non-small cell lung cancer and influence the progression of digestive system tumors (e.g., rectal and gastric cancer), its specific role and underlying mechanisms in bladder cancer remain largely unex-plored [15]. To date, only one study by Fujii et al. has reported on the relationship between AlkBH2 and bladder cancer, although the underlying mechanisms remain unclear [16].

Therefore, we hypothesize that AlkBH2 contributes to the progression of bladder cancer, potentially through the mechanism of augmenting tumor-associated inflam-mation. This study aims to explore the effects of AlkBH2 on the proliferation, colony formation, migration and invasion of bladder cancer cells, and explore that its mecha-nism might be achieved by activating the classic inflammatory pathway of NF-κB. Our findings may establish AlkBH2 as a new target for the treatment strategy of bladder cancer.

## Materials and methods

### Patients and tissue samples

A total of 58 paired tumor and adjacent non-tumor tissue samples were obtained from patients with bladder cancer who underwent transurethral bladder tumor resection or radical cystectomy at the Affiliated Hospital of Qinghai University from 1st August 2023–28th March 2024. All surgical specimens were histopathologically confirmed to be urothelial carcinoma by an experienced pathologist. The study was approved by the Ethics Committee of the Affiliated Hospital of Qinghai University (Study Identifier: P-SL-2023–463). Written informed consent was obtained from all participants prior to their inclusion in the study.

## Cell culture and transfection

Human umbilical vein endothelial cells (HUVEC), transitional cell carcinoma (TCCSUP) cells, and T24 cells (obtained from the Cell Bank of the Chinese Academy of Sciences, Shanghai, China) were authenticated by short tandem repeat (STR) profiling. These cell lines were maintained in Dulbecco's Modified Eagle's Medium (DMEM) supplemented with 10% heat-inactivated fetal bovine serum (FBS) and 1% penicillin-streptomycin, under conditions of 95% humidity and 5% $CO_2$ at 37°C. For hypoxia experiments, cells were cultured in an environment with 1% $O_2$. Lentiviral particles expressing AlkBH2 overexpression or shRNA constructs (GenePharma, China) were used to generate stable cell lines. The target sequences are provided in S1 Table. Cells ($3–5 \times 10^5$) were seeded in 6-well plates to achieve 60–70% confluency and transduced with lentivirus with green fluorescent protein (GFP). The transduction groups included control group, AlkBH2 overexpression group, and AlkBH2 knockdown group with a multiplicity of infection (MOI) of 10 (virus titer: $2 \times 10^8$ TU/mL). After 6 hours, the medium was replaced with fresh DMEM. Transfection efficiency was evaluated 48 hours post-transduction by fluorescence microscopy, quantifying the percentage of GFP-positive cells, compared with the number of cells under white light.

## Histological analysis

Bladder tissue samples were fixed in 4% paraformaldehyde for 12 hours and sectioned at a thickness of 5 μm. Hematoxylin and eosin (H&E) staining was performed as follows: slides were deparaffinized, stained with eosin for 5 seconds, and then immersed in hematoxylin for 1 minute. High-resolution images were captured using a Leica DM500 microscope (Leica Microsystems, Solms, Germany).

## Immunofluorescence (IF)

Following deparaffinization, tissue sections were permeabilized with 0.5% Triton X-100 for 20 minutes and blocked with goat serum for 1 hour at room temperature. Sections were then incubated with primary antibodies (S1 Table) overnight at 4°C, followed by incubation with fluorescein- or rhodamine-conjugated goat anti-mouse/rabbit IgG secondary antibodies for 1 hour. Nuclei were counterstained with DAPI for 5 minutes before examination under a microscope (Tissue Gnostics, Austria).

## Q-PCR

Total RNA was isolated from cell samples using the GeneJET RNA Purification Kit (Thermo Fisher Scientific, Waltham, MA, USA). cDNA was synthesized from the extracted RNA using the Goldenstar RT6 cDNA Synthesis Kit (Tsingke Biotechnology, Beijing, China). Quantitative real-time PCR (qPCR) was performed using the Master qPCR Mix (Tsingke Biotechnology, Beijing, China) on a CFX96 Touch Q-PCR Detection System (Bio-Rad). Primers were synthesized by Tsingke Biotechnology, and Actin was used as an internal control to normalize gene expression levels. Relative gene expression was calculated using the $2^{-\Delta\Delta CT}$ method. Primer sequences are listed in S1 Table.

## Western Blot

Tissues or cells were homogenized in protein lysis buffer using ultrasonic equipment (KeyGEN, Jiangsu, China) and incubated on ice for 10 minutes. The lysates were subjected to sodium dodecyl sulfate-polyacrylamide gel electrophoresis (SDS-PAGE) and then transferred to polyvinylidene difluoride (PVDF) membranes. The membranes were blocked with 3% bovine serum albumin (BSA) in PBS for 1 hour at room temperature (S2 Fig). They were then incubated with primary antibodies overnight at 4°C, followed by incubation with horseradish peroxidase (HRP)-conjugated secondary antibodies (1:10000 dilution) for 1 hour at room temperature. Immunoreactive bands were visualized using chemiluminescent reagents. Details regarding the primary antibodies used are provided in S1 Table.

## Real-time cell analysis

After the "background measurement" protocol was executed using the integrated software to record the baseline impedance for each well, three groups of TCCSUP and T24 cell lines in the logarithmic growth phase were detached using 0.25% trypsin-EDTA and resuspended in complete medium. Following background measurement, the E-Plate was briefly removed from the analyzer. Based on optimization from preliminary experiments, cells were seeded at a density of 2,000 cells per well in a final volume of 100 µL complete medium per well. After seeding, the plate was left undisturbed at room temperature for 30 minutes to facilitate uniform cell settling. The plate was then returned to the RTCA analyzer (Agilent Technologies, USA), and continuous impedance monitoring was initiated with measurements taken every 15 minutes for the first 6–8 hours to track cell attachment, followed by hourly recordings for a total duration of at least 50 hours.

## Cell cycling

The proportion of cells in different phases of the cell cycle was determined by flow cytometry. Cells were fixed in 70% ethanol, stained with propidium iodide (PI) and RNase A, and analyzed to determine the distribution of cells in G0/G1, S, and G2/M phases based on DNA content using a cell cycle kit (Beyotime, Shanghai, China).

## Colony formation assay

Cells ($2 \times 10^3$ per well) were seeded into 6-well plates and cultured for 7 days to allow colony formation. Visible colonies were fixed with 95% ethanol for 30 min, stained with 1% crystal violet for 15 min, and then photographed and counted.

## Wound-healing assay

Cells ($9 \times 10^5$) were plated in 6-well plates and allowed to reach 100% confluence. A linear wound was created by scratching the monolayer with a 10 µL pipette tip. Cells were cultured with 10 µg/mL mitomycin C (MCE, New Jersey, USA) at 37°C in 5% $CO_2$, and images were captured after 36 hours to assess the rate of wound closure.

## Migration and invasion assays

For the cell migration assay, $3 \times 10^4$ cells were seeded in the upper chamber of a Transwell plate (Corning, USA), while 500 µL of culture medium supplemented with 20% FBS was added to the lower chamber as a chemoattractant. For the invasion assay, the Transwell chambers were pre-coated with Matrigel (Corning, USA) before seeding the cells. Cells were incubated at 37°C for 24 or 48 hours. Cells in the upper chamber were removed, and those that had migrated or invaded were fixed with 4% paraformaldehyde, stained with crystal violet, and quantified using ImageJ software.

## Endothelial cell tube formation assay

HUVEC cells were subjected to starvation by replacing the complete culture medium with DMEM containing 0.2% FBS for 24 hours. Growth Factor Reduced Matrigel (Corning, USA) was thawed overnight at 4°C. Using pre-chilled pipette tips, 20 µL of Matrigel was carefully added to each well of a 24-well plate. The plate was then incubated at 37°C for 30−45 minutes to allow polymerization. Starved HUVECs were trypsinized and resuspended to form a single-cell suspension. The assay medium was prepared by mixing normalized conditioned medium, which required that the total number of uniform cells from each bladder cancer cell group is $10^6$ cultured for 48 hours, with fresh endothelial cell growth medium (EGM-2), supplemented with 10% FBS, to create the final treatment media. HUVECs were seeded onto the polymerized Matrigel at a density of $5 \times 10^4$ cells per well in 500 µL of the respective treatment or control media. The plate was immediately placed in a humidified incubator at 37°C with 5% $CO_2$. Images were captured at 8 hours to observe tube formation. Tube formation was quantified using image analysis software (ImageJ with the angiogenesis analyzer plugin) by measuring key parameters such as total tube length, number of master segments, number of branching points, and total mesh area per field of view.

### Enzyme-linked immunosorbent assays (ELISA)

For each experimental group, six samples were independently prepared for protein extraction and concentration determination. Microplates were coated with capture antibody (5 µg/mL in PBS, 100 µL per well) and incubated overnight at 4°C. After washing three times with PBS containing 0.05% Tween-20, the plates were blocked with 200 µL of 5% BSA in PBS for 2 hours at room temperature. Standards and samples, diluted in 1% BSA/PBS, were added to the wells (100 µL per well) and incubated for 2 hours at room temperature. Following another wash, biotinylated detection antibody was added and incubated for 1 hour at room temperature, followed by Streptavidin-HRP (1:5000 dilution, 100 µL per well) for 30 minutes at room temperature. After the final wash, TMB substrate (100 µL per well) was added and allowed to develop in the dark at room temperature for 15 minutes. The reaction was stopped with 1M $H_2SO_4$ (50 µL per well), and absorbance was measured at 450 nm using a microplate reader (Thermo Scientific, Waltham, MA, USA). Detailed information on the specific ELISA kits used is provided in S1 Table.

### Proteomic analysis

For each group, three samples were sequenced repeatedly for protein extraction and determination of their concentrations, followed by protein digestion and peptide segment desalination. Before the mass spectrometry sample injection, each sample was mixed at a volume ratio of iRT: sample to be measured = 1:20 by volume, serving as an internal standard. Equal amounts of peptide segments were taken from all the enzymatically digested samples and separated using the EASY-nLC 1200 liquid chromatography. The mobile phase A was 0.1% FA aqueous solution, and the mobile phase B was 0.1% FA ACN. Gradient elution conditions: 0–20 minutes, 5–22% B; 20~24 min, 22–37% B; 24~27 min, 37–80% B; 27–30 minutes, 80% B. The peptide segments were separated by the ultra-high performance liquid chromatography system and then injected into the timsTOF Pro mass spectrometer (Bruker) for analysis. The mass spectrometry conditions were as follows: capillary voltage 1.4 KV, drying gas temperature 180°C, drying gas flow rate 3.0 L/min, mass spectrometry scanning range 100–1700 m/z, ion drift range 0.7–1.3 Vs/cm2, collision energy range 20–59 eV. The raw data of DIA was processed using Spectronaut Pulsar™ 18.4 (Biognosys, Switzerland) software. The mass spectrometry search parameters were precursor mass value threshold 0.01, protein mass value threshold 0.01, fixed modification as Carbamidomethyl(C), variable modifications as Oxidation(M) and Acetyl(N-term), maximum missed cleavage site 2. The differentially expressed proteins need to meet $p < 0.05$ and FC > 1.2 or FC < 1/1.2. When P < 0.05 and FC > 1.2, it is a significantly up-regulated protein; when $p < 0.05$ and FC < 1/1.2, it is a significantly down-regulated protein. The data was imported into the corresponding database for analysis.

### Statistical analysis

SPSS v22.0 software was used for statistical analysis. The data were quantified as the mean±SD. One-way analysis of variance (ANOVA) was applied in more than three groups. A t-test was used to compare the mean values of the two groups of samples. The rates were analyzed using the rank test, and $p < 0.05$ indicated statistical significance.

## Results

### High expression of AlkBH2 in bladder cancer

Although AlkBH3 is well-documented for its high expression in urological tumors, its homolog, AlkBH2, has received comparatively less attention [17]. To address this gap, we measured the expression levels of AlkBH2 in bladder cancer tissues relative to para-carcinoma tissues. A total of 58 bladder cancer cases from the Affiliated Hospital of Qinghai University were confirmed by HE staining. Histopathological examination revealed that bladder cancer tissues exhibited disorganized cellular architecture, loss of polarity, and disruption of the stratified arrangement from the basal to the superficial layers. Cells displayed marked pleomorphism, characterized by variations in size and shape, enlarged nuclei, an increased

nuclear-to-cytoplasmic ratio, anisokaryosis (nuclear size variation), and irregular nuclear contours. Pathological mitotic figures were also observed (Fig 1A). IF analysis demonstrated that AlkBH2 was significantly upregulated in bladder cancer tissues (Fig 1B, C). Consistent with this, both Q-PCR and Western blot analyses confirmed that the mRNA and protein levels of AlkBH2 were significantly higher in bladder cancer tissues compared to normal tissues. These findings suggest that high AlkBH2 expression is prevalent in bladder cancer patients and may play a crucial role in the malignant progression of the disease.

## Models of overexpression and silencing in bladder cancer

To elucidate the functional role of AlkBH2 in bladder cancer, we utilized T24 and TCCSUP bladder cancer cell lines. Lentiviral vectors were employed to establish stable models of AlkBH2 overexpression and silencing in these cell lines. Immunofluorescence microscopy revealed that the transfection efficiency in both T24 and TCCSUP cells exceeded 85% (Fig 2A-C). To validate the success of these models, Q-PCR and Western blot analyses were performed to quantify the expression of AlkBH2 RNA and protein in the control, overexpression, and silencing groups. Compared to the control, AlkBH2 expression was significantly upregulated in the overexpression group, while it was markedly downregulated in the silencing group (Fig 2D-I). These results confirm the successful establishment of stable models for further exploration of AlkBH2's effects.

## AlkBH2 enhances the proliferation of bladder cancer

Building upon the successful establishment of these models, we investigated the impact of AlkBH2 on tumor proliferation in bladder cancer. Real-time cell analysis demonstrated that AlkBH2 overexpression significantly accelerated cell proliferation, whereas AlkBH2 knockdown had the opposite effect in both T24 and TCCSUP cells (Fig 3A). Additionally, colony formation assays showed that AlkBH2 overexpression promoted colony formation, while AlkBH2 knockdown inhibited it (Fig 3B, C). Flow cytometry analysis of the cell cycle distribution revealed that AlkBH2 overexpression increased the proportion of cells in the S and M phases, indicative of enhanced cell division, while AlkBH2 knockdown caused a significant G1 phase arrest in both T24 and TCCSUP cells (Fig 3D-F). Collectively, these findings indicate that AlkBH2 promotes the growth of bladder cancer cells by facilitating cell cycle progression, while its depletion impairs proliferative capacity.

## AlkBH2 promotes migration and invasion in bladder cancer

In addition to its role in proliferation, the ipact of AlkBH2 on other aspects of tumor heterogeneity warrants further examination. Wound-healing assays demonstrated that overexpression of AlkBH2 significantly increased the migratory capacity of bladder cancer cells, whereas knockdown of AlkBH2 markedly inhibited cell migration compared to the control group (Fig 4A, B). To elucidate the role of AlkBH2 in angiogenesis, we performed tube formation assays, which revealed that AlkBH2 overexpression substantially accelerated the formation of capillary-like structures by endothelial cells, while AlkBH2 knockdown suppressed this process (Fig 4C, D). Transwell assays further confirmed that overexpression of AlkBH2 enhanced the invasive potential of bladder cancer cells (Fig 4E, F). Together, these findings demonstrate that AlkBH2 plays a crucial role in promoting proliferation, migration, and invasion in bladder cancer, suggesting its potential as a key driver in tumor progression.

## AlkBH2 downregulation inhibits inflammation in bladder cancer

While the precise mechanisms underlying the role of AlkBH2 in bladder cancer development remain unclear, our hypothesis posits that AlkBH2 may modulate tumor-associated inflammation. The dysregulation of DNA repair factors can compromise genomic integrity and exacerbate DNA damage, which in turn can trigger a potent inflammatory response [18]. Given that tumor-related inflammation is a fundamental characteristic of cancer, influencing various stages of tumor progression

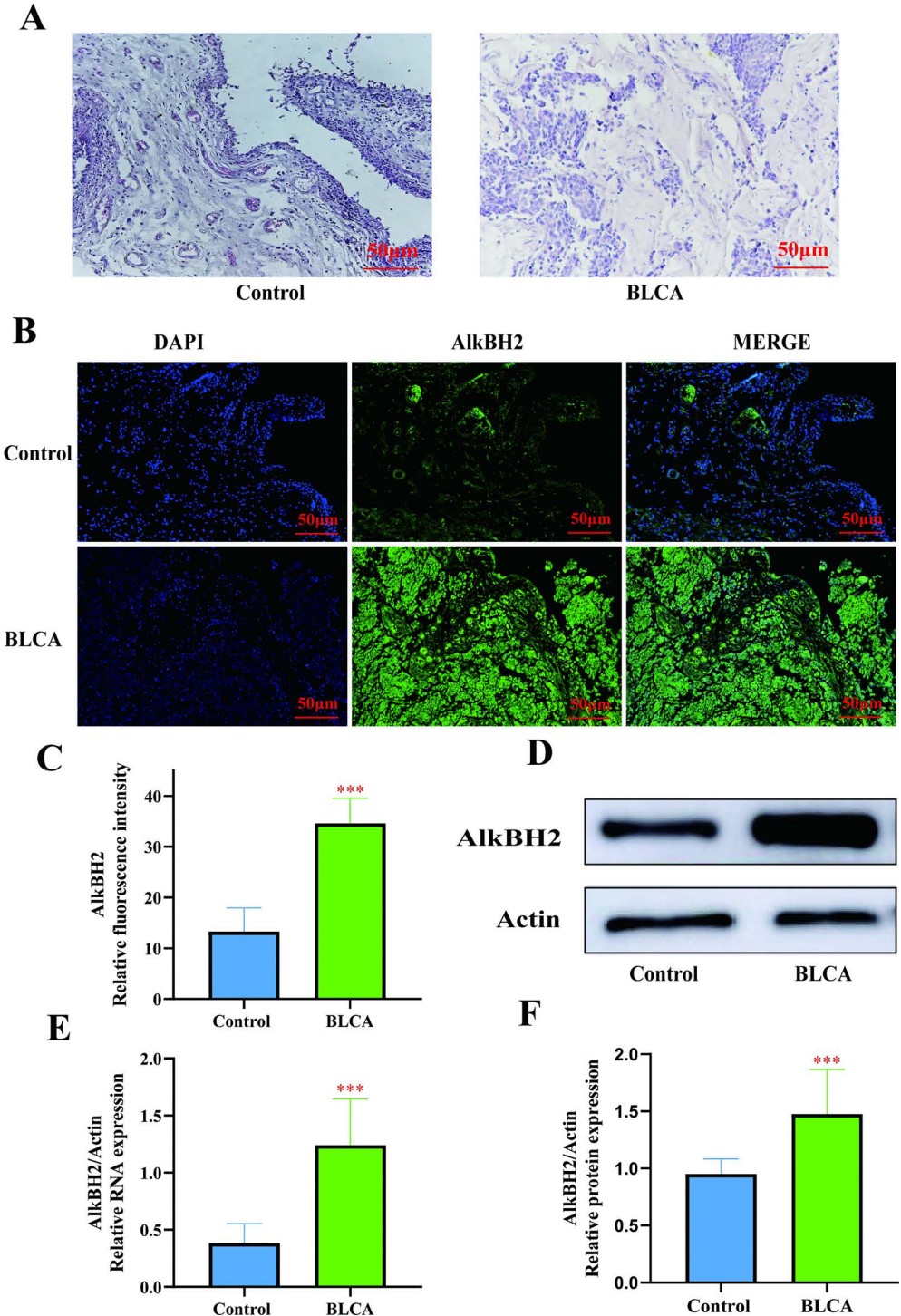

**Fig 1. Expression of AlkBH2 in bladder cancer. (A)** HE stained images of bladder cancer and adjacent normal tissues from patients. Scale bar, 50 μm. **(B)** Representative immunohistochemical images showing AlkBH2 expression in a bladder cancer tissue microarray. Scale bar, 50 μm. **(C)** Quantitative analysis of relative fluorescence intensity of AlkBH2, normalized to the hypoxic group (n=6). **(D)** Representative Western blot image of AlkBH2 in bladder cancer tissue. **(E)** Analysis of AlkBH2 mRNA expression levels in normal and bladder cancer tissues (n=6). **(F)** Analysis of AlkBH2 protein expression levels in normal and bladder cancer tissues (n=6). Data are shown as mean±SD. Statistical significance was assessed using a two-tailed unpaired Student's t-test. *** $P < 0.001$, ** $P < 0.01$, * $P < 0.05$.

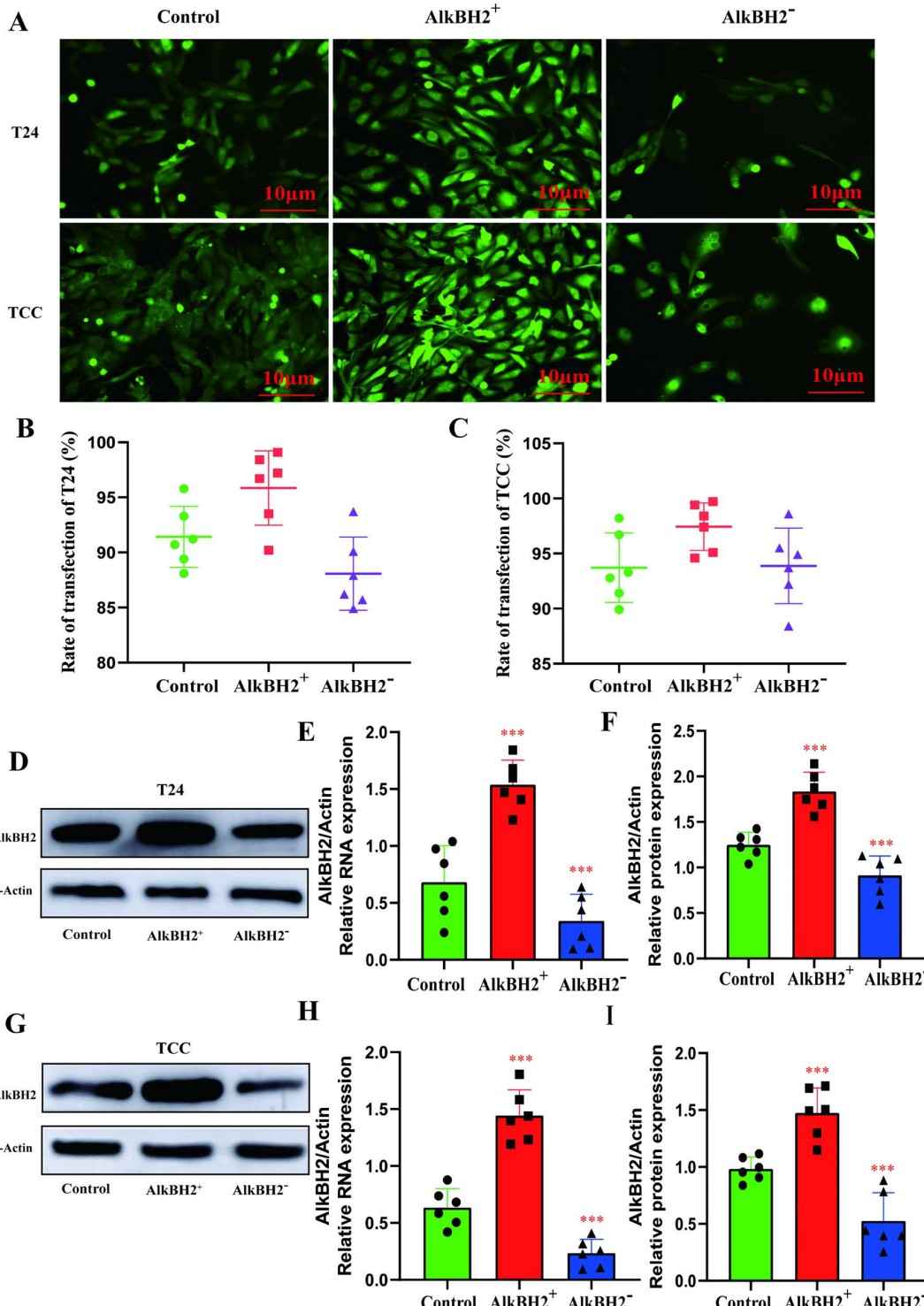

**Fig 2. Overexpression and knockdown of AlkBH2 in bladder cancer cell lines. (A)** Immunofluorescence microscopy images of T24 and TCCSUP cells transfected with lentiviruses. Scale bar, 50 µm. **(B, C)** Transfection efficiency was quantified in T24 and TCCSUP cells, respectively (n = 3). **(D)** Representative Western blot image of AlkBH2 in T24 cells. **(E, F)** Analysis of AlkBH2 mRNA and protein expression levels in T24 cells, compared to the control group (n = 3). **(G)** Representative Western blot image of AlkBH2 in TCCSUP cells. **(H, I)** Analysis of AlkBH2 mRNA and protein expression levels in TCCSUP cells, compared to the control group (n = 3). Data are shown as mean ± SD. Statistical significance was assessed using a two-tailed unpaired Student's t-test. *** $P < 0.001$, ** $P < 0.01$, * $P < 0.05$.

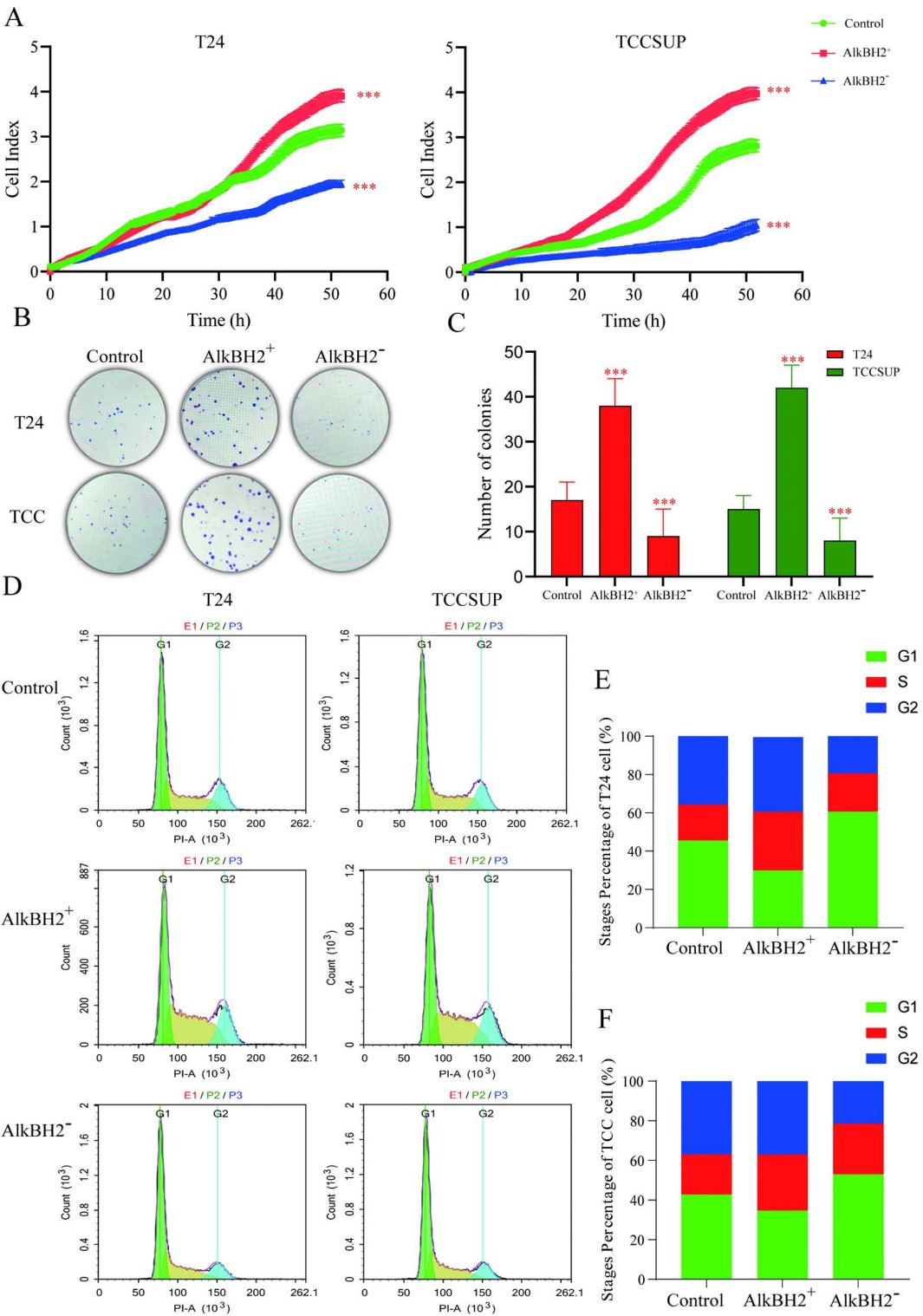

**Fig 3. AlkBH2 promotes the proliferation of bladder cancer cells. (A)** CCK-8 assay to quantify cell proliferation (n = 6). **(B)** Colony formation assay to assess cell proliferation in bladder cancer cells. **(C)** Quantification of colony numbers at low magnification (n = 3). **(D)** Flow cytometric analysis of the cell cycle in T24 and TCCSUP cells. **(E, F)** Analysis of the distribution of T24 and TCCSUP cells in the G1, S, and G2 phases, respectively (n = 3). Data are shown as mean ± SD. Statistical significance was assessed using a two-tailed unpaired Student's t-test. *** $P < 0.001$, ** $P < 0.01$, * $P < 0.05$.

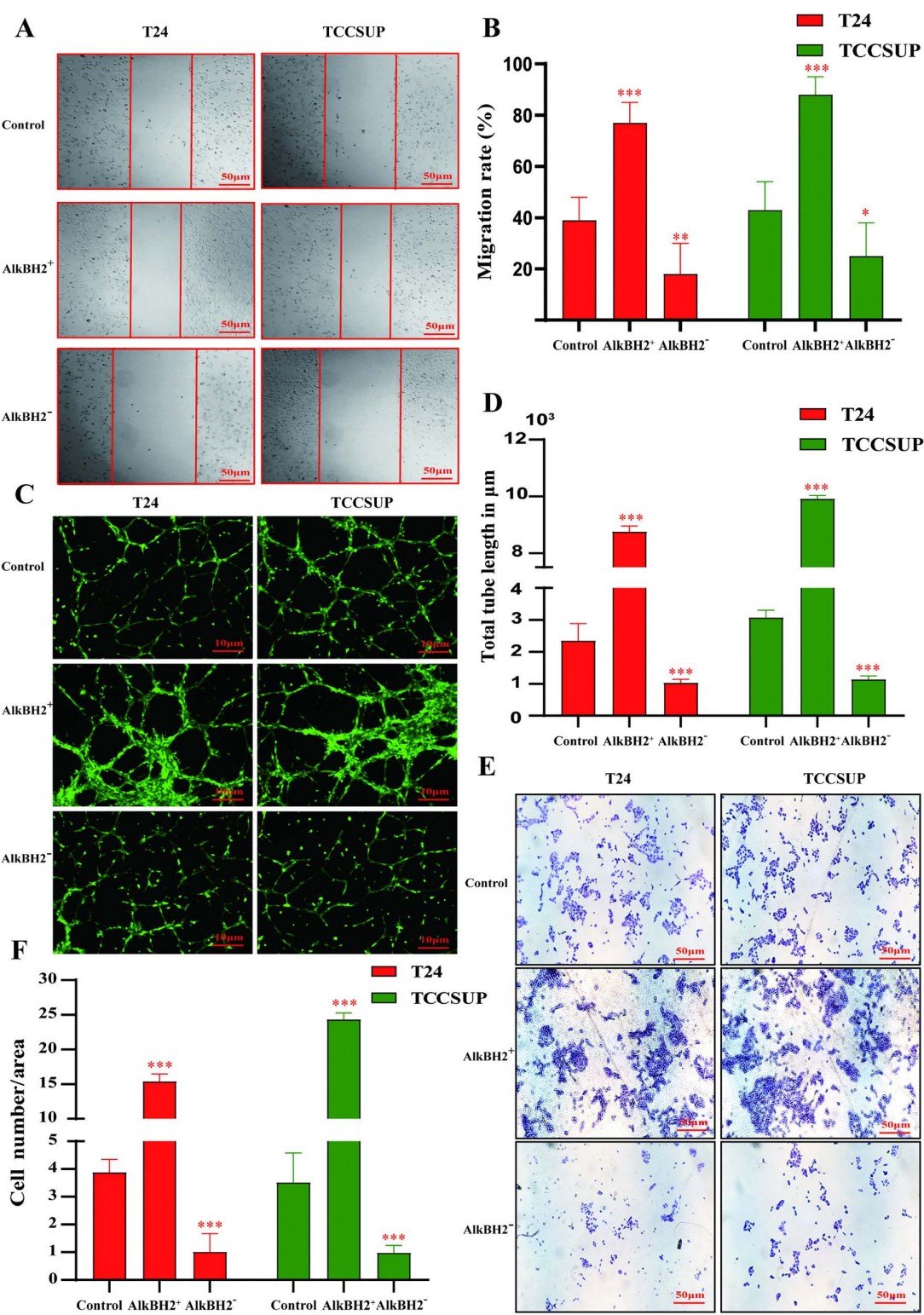

**Fig 4. AlkBH2 enhances the migration and invasion in bladder cancer. (A)** Wound healing assay to evaluate the migratory potential of cells. Scale bar, 50 μm. **(B)** Quantification of wound closure rates, compared to the control group (n = 3). **(C)** Tube formation assay to assess angiogenic potential under immunofluorescence microscopy. Scale bar, 10 μm. **(D)** Quantification of total tube length (n = 3). **(E)** Transwell assay to evaluate cell invasion. Scale bar, 50 μm. **(F)** Quantification of invading cells (n = 3). Data are shown as mean ± SD. Statistical significance was assessed using a two-tailed unpaired Student's t-test. *** $P < 0.001$, ** $P < 0.01$, * $P < 0.05$.

[19], we investigated the levels of inflammatory cytokines in T24 and TCCSUP cells. Enzyme-linked immunosorbent assays (ELISAs) showed that overexpression of AlkBH2 upregulated proinflammatory factors such as IL-1β, TNF-α, IL-12, and IL-17, indicating that AlkBH2 activates inflammation to promote cancer development (Fig 5A-D). This suggests that AlkBH2 fosters a pro-tumorigenic inflammatory environment. Conversely, knockdown of AlkBH2 upregulated anti-inflammatory factors such as IL-10, IL-4, TGF-β, and IL-38, thereby suppressing tumor growth (Fig 5E-H). Collectively, these results suggest that AlkBH2 contributes to bladder cancer progression, at least in part, by skewing the tumor micro-environment toward a pro-inflammatory state.

### AlkbH2 upregulates inflammation in bladder cancer via activation of NF-κB and suppression of NRF2/HO-1

To elucidate the molecular mechanisms underlying AlkBH2's influence on bladder cancer, we conducted a proteomic analysis of bladder cancer cells treated with AlkBH2. Volcano plots revealed a significant number of differentially expressed proteins upon AlkBH2 treatment in both control and AlkBH2 treated samples (Fig 6A, B), with the heatmap representation further illustrating the distinct protein expression profiles between the two groups (Fig 6C, D). KEGG pathway enrichment analysis (Fig 6E) highlighted the enrichment of the NF-κB signaling pathway. Further investigation of the NF-κB signaling pathway map (Fig 6F) revealed alterations in the expression levels of key components within this pathway, suggesting that AlkBH2 may modulate inflammatory responses in bladder cancer cells via the NF-κB signaling pathway. The role of nuclear factor kappa-light-chain-enhancer of activated B cells (NF-κB), a key transcription factor involved in regulating inflammation [20], oxidative stress [21], immune responses [22], and tumorigenesis [23]. In unstimulated cells, NF-κB dimers are sequestered in the cytoplasm by inhibitory proteins (IκBα, IκBβ, IκBε) [24]. Upon activation by various stimuli, IκB proteins are phosphorylated, leading to their degradation and the subsequent nuclear translocation of NF-κB [25].

Our results showed that overexpression of AlkBH2 significantly increased the phosphorylation and nuclear translocation of NF-κB, while knockdown of AlkBH2 suppressed these processes (Fig 7A-D). In addition, we focused on the its upstream signaling pathway, which plays a pivotal role in anti-inflammatory and antioxidant responses [26]. Overexpression of AlkBH2 downregulated the expression of NRF2 (Nuclear factor erythroid-2-related factor 2) and HO-1 (Heme oxygenase-1), while knockdown of AlkBH2 restored the activity of the NRF2/HO-1 pathway (Fig 7E-H). As an upstream regulator of NF-κB, the exploration of the NRF2/HO-1 signaling pathway further revealed that AlkBH2 significantly upregulated the inflammation in bladder cancer, thereby promoting the development of the tumor.

In order to investigate the location of NF-κB in cell, we extracted the nuclear proteins, using Lamin B1 as the internal control (Fig 7I-L), and immunofluorescence staining confirmed that overexpression of AlkBH2 enhanced NF-κB nuclear localization, thereby activating its downstream inflammatory responses and promoting bladder cancer development (Fig 8A-D). In summary, AlkBH2 facilitates bladder cancer progression by upregulating inflammation through the activation of NF-κB and the suppression of the NRF2/HO-1 pathway.

## Discussion

AlkBH2, a key dioxygenase, repairs alkylation damage in genomic DNA and is overexpressed in many cancers [27]. Among the human homologs, only AlkBH2 and AlkBH3 have demonstrated repair activities analogous to the bacterial AlkB enzyme from *Escherichia coli* [28]. AlkBH2 functions as a housekeeping enzyme, safeguarding the mammalian genome from 1-methyladenine (1-meA) damage by facilitating the repair of double-stranded DNA (dsDNA) [29]. Although the role of AlkBH3 in tumorigenesis has been extensively investigated, relatively little is known about AlkBH2. This study aims to elucidate the role of AlkBH2 in bladder cancer, with our findings indicating that silencing AlkBH2 effectively inhibits tumor progression by suppressing cellular proliferation, migration, and invasion. These results suggest that AlkBH2 exerts a significant influence on bladder cancer development and represents a potential therapeutic target.

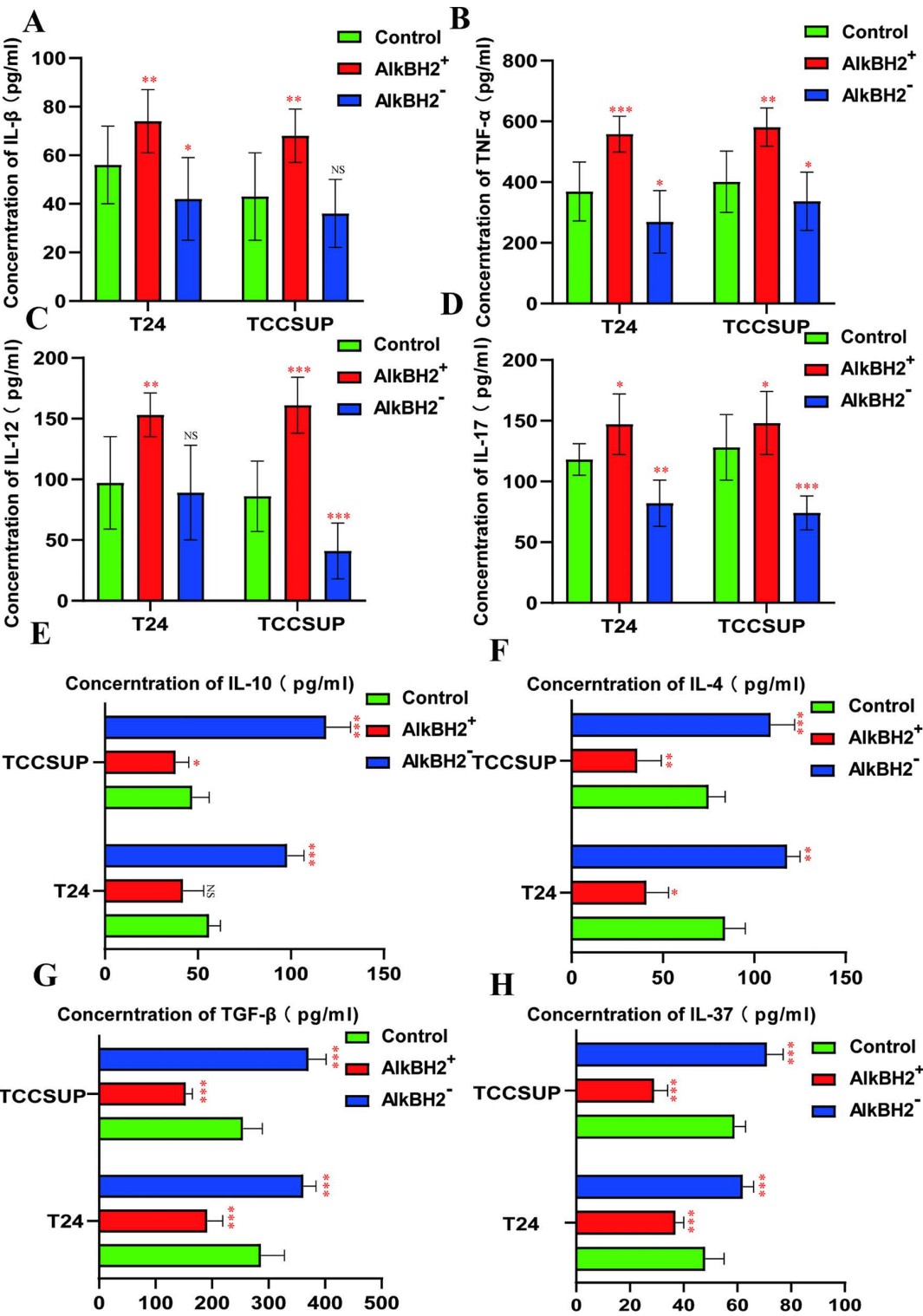

**Fig 5. AlkBH2 upregulates inflammatory factors in bladder cancer. (A-D)** ELISA to measure pro-inflammatory cytokine levels in T24 and TCCSUP cells (n = 6). **(E-H)** ELISA to measure anti-inflammatory cytokine levels in T24 and TCCSUP cells (n = 6). Data are shown as mean ± SD. Statistical significance was assessed using a two-tailed unpaired Student's t-test. *** $P < 0.001$, ** $P < 0.01$, * $P < 0.05$.

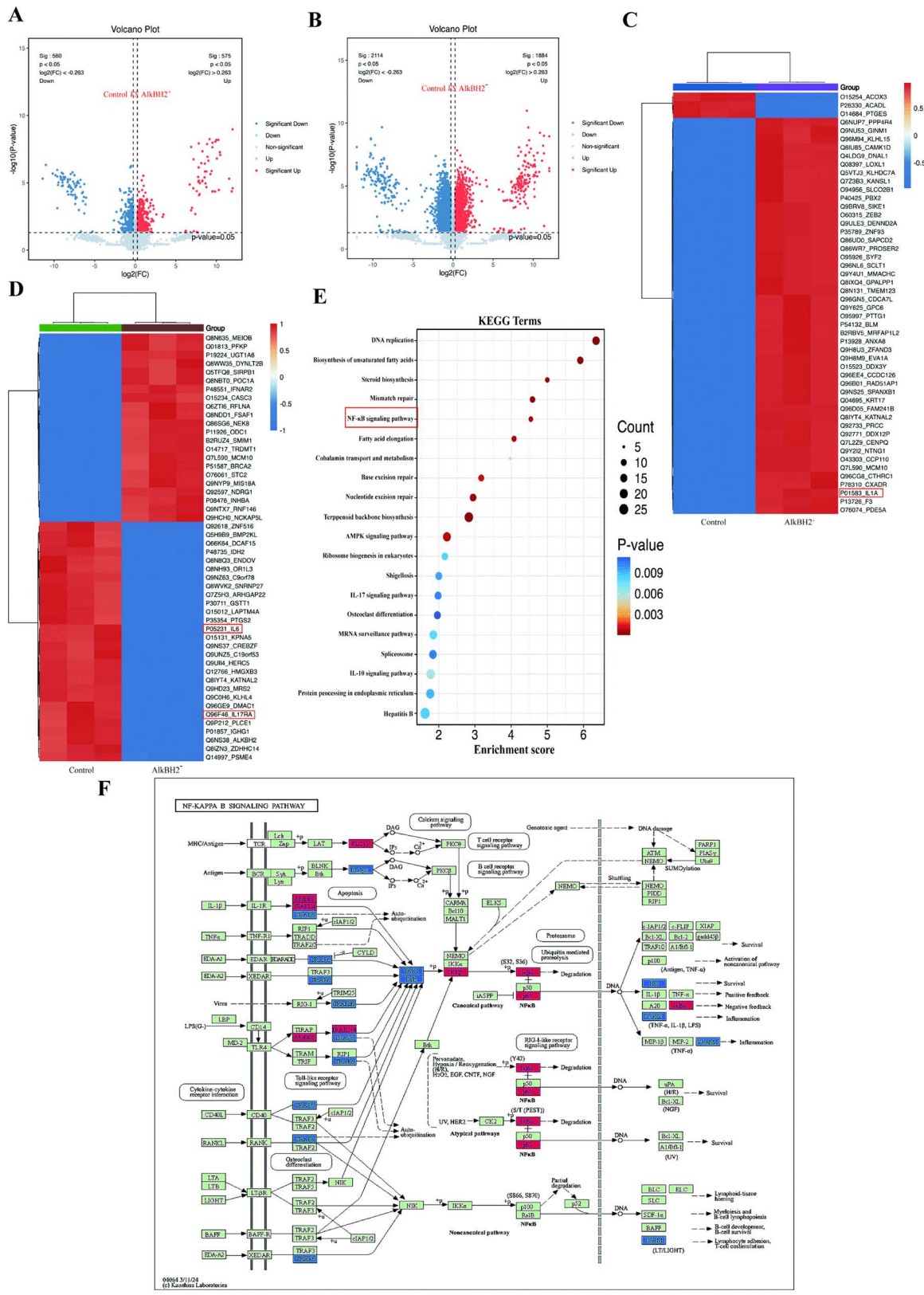

**Fig 6. Proteomic Analysis Reveals the Involvement of AlkBH2 in Bladder Cancer via the NF-κB Signaling Pathway. (A, B)** Volcano plots depicting differentially expressed proteins in bladder cancer cells treated with AlkBH2 compared to the control group. **(C, D)** Heatmaps representing the

expression patterns of differentially expressed proteins between the control and AlkBH2-treated groups. **(E)** KEGG pathway enrichment analysis of the differentially expressed proteins. The red box highlights the NF-κB signaling pathway. **(F)** Schematic representation of the NF-κB signaling pathway, with proteins showing altered expression levels after AlkBH2 treatment highlighted in blue (downregulated) or red (upregulated). The source of data from KEGG data: https://www.kegg.jp/keggbin/search_pathway_text?map=map&keyword=NF-KB&mode=1&viewImage=true.

However, the mechanism by which AlkBH2 promotes the development of bladder cancer has not yet been elucidated. Based on the proteomics results, we discovered the role of AlkBH2 in the inflammation of bladder cancer. The relationship between inflammation and cancer has been recognized since the 19th century, with inflammation identified as a potential etiological factor [30]. The link between chronic inflammation and cancer is well-established, with persistent inflammation contributing to tumor initiation and progression across various malignancies [31]. The deficiency of tumor suppressor factors impairs DNA repair mechanisms, exacerbating DNA damage and triggering an inflammatory cascade [32]. Within the tumor microenvironment, a complex interplay of immune cells, cytokines, and chemokines drives a pro-tumorigenic inflammatory response [33]. This response can directly affect malignant cells—for instance, by inducing epithelial-mesenchymal transition—and indirectly support tumor growth through angiogenesis and tissue remodeling, solidifying inflammation as a hallmark of cancer [34]. Therefore, we hypothesize that AlkBH2 may promote bladder cancer by facilitating inflammation in tumor. We found that the expression levels of pro-inflammatory cytokines, including IL-1β, TNF-α, IL-12, and IL-17, were significantly upregulated in cells overexpressing AlkBH2, indicating that AlkBH2 activates inflammation in bladder cancer. Conversely, the levels of anti-inflammatory cytokines, such as IL-10, IL-4, TGF-β, and IL-38, were downregulated. These findings suggest that one of the mechanisms by which AlkBH2 promotes bladder cancer progression involves the activation of inflammation, potentially increasing the heterogeneity of bladder cancer.

To elucidate the mechanisms by which AlkBH2 promotes inflammation in bladder cancer, we focused on nuclear factor kappa B (NF-κB) by proteomics enrichment analysis, a family of transcription factors involved in regulating a broad spectrum of biological responses [35]. NF-κB is well-known for its role in immune responses and inflammation, but recent evidence supports its significant involvement in oncogenesis [36]. Our study demonstrates that AlkBH2 is a novel upstream regulator of this pathway in bladder cancer. Specifically, we found that AlkBH2 overexpression enhances NF-κB phosphorylation, promotes its nuclear translocation—as confirmed by nuclear fractionation and immunofluorescence—and amplifies its downstream transcriptional activity. Conversely, AlkBH2 knockdown suppresses these events. This positions AlkBH2 as a direct activator of the NF-κB signaling cascade, mechanistically explaining its pro-inflammatory role in the tumor microenvironment. These findings integrate AlkBH2 into the established paradigm of NF-κB-driven tumor progression, while highlighting its novelty as an epigenetic modifier regulating this pathway. Therapeutically, this suggests that targeting AlkBH2 could disrupt a specific node in NF-κB activation, offering a potential alternative to broader NF-κB inhibitors. A limitation of this study is that the precise molecular step by which AlkBH2 induces NF-κB phosphorylation remains to be determined. Future work should identify the specific kinases involved and validate these interactions in vivo. Ultimately, linking AlkBH2 to NF-κB activation provides a clear mechanism for its pro-tumorigenic effects and underscores its potential as a therapeutic target in bladder cancer.

Notably, we further elucidated an upstream mechanism by which AlkBH2 regulates this process. NRF2/HO-1 signal pathway, as the upstream signaling pathway of NF-κB, is also another mechanism we should be discussed. Our results indicate that AlkBH2 suppresses the NRF2/HO-1 signaling pathway [37], a critical cellular defense system against oxidative stress and inflammation [38]. Given the documented protective and anti-inflammatory roles of NRF2/HO-1 in various cancers, its inhibition by AlkBH2 provides a novel explanatory mechanism for the observed upregulation of NF-κB and subsequent inflammatory activation [39]. This suggests AlkBH2 may promote bladder cancer progression not only by directly enhancing NF-κB activity but also by disabling a key anti-inflammatory and antioxidant pathway. While this study identifies AlkBH2 as a regulator bridging NRF2/HO-1 suppression to NF-κB activation, several limitations must be

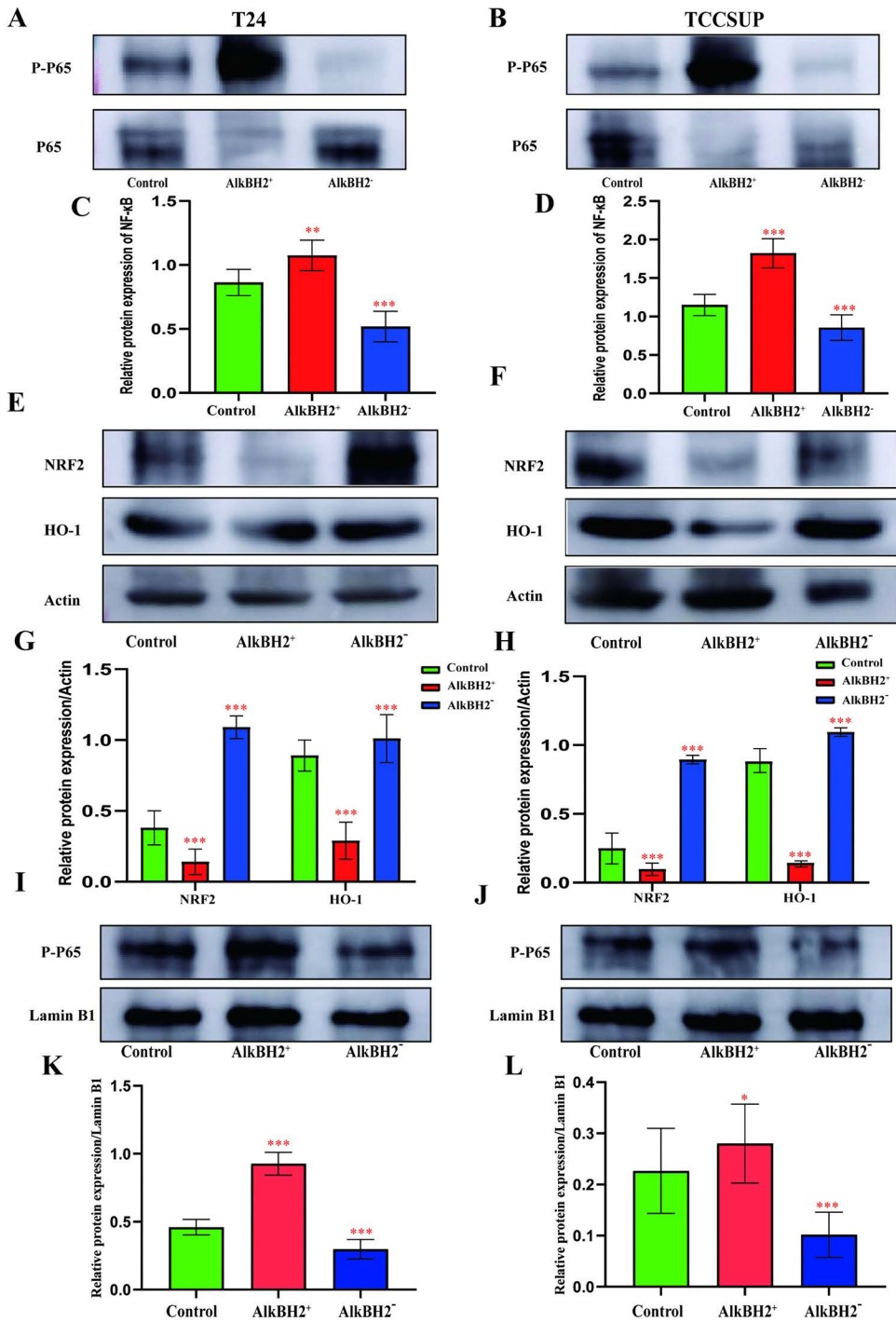

**Fig 7. Mechanism of AlkbH2-mediated inflammation upregulation. (A, B)** Representative Western blot images of P-P65 and P65 expression in T24 and TCCSUP cells. **(C, D)** Densitometric analysis of P-P65 and P65 expression levels in the three groups, compared to the control (n = 3). **(E, F)** Representative Western blot images of NRF2 and HO-1 expression in T24 and TCCSUP cells. **(G, H)** Densitometric analysis of NRF2 and HO-1 expression levels in the three groups, compared to the control (n = 3). **(I, J)** Representative Western blot images of P-P65 expression in nucleus of T24 and TCCSUP cells. Lamin B1 was used as a loading control. **(K, L)** Densitometric analysis of P-P65 expression levels of nucleus in the three groups, compared to the control (n = 3). Data are shown as mean ± SD. Statistical significance was assessed using a two-tailed unpaired Student's t-test. *** $P < 0.001$, ** $P < 0.01$, * $P < 0.05$.

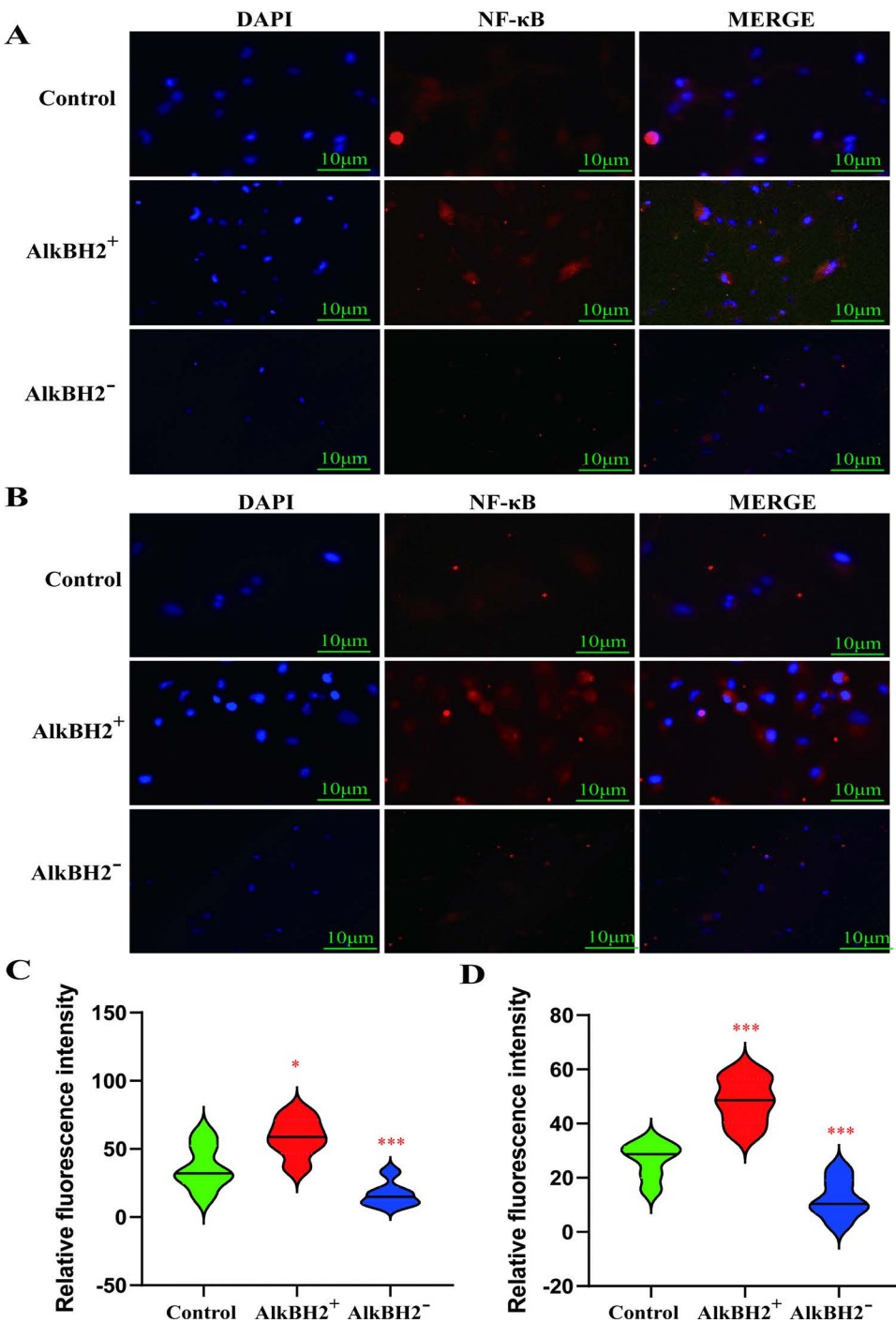

**Fig 8. AlkbH2 upregulates inflammation in bladder cancer via nuclear translocation of NF-κB. (A, B)** Representative immunohistochemical images of NF-κB signaling pathway expression in bladder cancer tissue microarrays from T24 and TCCSUP cells. **(C, D)** Quantification of relative fluorescence intensity of NF-κB, compared to the control (n = 3). Data are shown as mean ± SD. Statistical significance was assessed using a two-tailed unpaired Student's t-test. *** $P < 0.001$, ** $P < 0.01$, * $P < 0.05$.

acknowledged. The precise molecular step at which AlkBH2 inhibits the NRF2/HO-1 axis remains to be defined. Furthermore, the functional consequences of this inhibition, such as its specific contribution to angiogenesis or metastasis, require further investigation. Nonetheless, by positioning AlkBH2 as a modulator of both pathways, our work highlights a potential therapeutic target for disrupting inflammatory signaling in bladder cancer.

In summary, our study investigated the effects of AlkBH2 on bladder cancer cell proliferation, colony formation, migration, and invasion. We found that AlkBH2 promotes bladder cancer progression by upregulating inflammation through increased NF-κB phosphorylation and nuclear translocation. This suggests a novel mechanism by which AlkBH2 influences bladder cancer development and offers potential therapeutic strategies. While this work establishes a foundational link between AlkBH2 and inflammatory signaling in bladder cancer, further investigation is required to elucidate the precise molecular mechanisms connecting AlkBH2 activity directly to the NF-κB pathway and to validate these findings in vivo. Future research will focus on delineating this relationship and exploring the translational potential of targeting AlkBH2 in bladder cancer treatment.

## Supporting information

**S1 Table. List of key reagents information. (A)** The manufacturer's name and product number of the key reagents of the ELISA kit. **(B)** The primer sequence of AlkBH2 mRNA and the silencing sequence of AlkBH2. **(C)** The dilution concentration of the antibody reagent, as well as the manufacturer's name and product number. **(D)** The manufacturer's name and product number of relevant agents.
(DOCX)

**S2 Fig. The original blot and gel.**
(TIF)

## Author contributions

**Conceptualization:** Zhangjie Yang, Guojun Chen.

**Data curation:** Ziyang Qiang.

**Formal analysis:** Wenhao Xie.

**Funding acquisition:** Guojun Chen.

**Methodology:** Zhangjie Yang, Jinhu Ma.

**Resources:** Wenhao Xie.

**Software:** Jinhu Ma.

**Supervision:** Wenhao Xie, Guojun Chen.

**Validation:** Ziyang Qiang, Liang Jiao.

**Visualization:** Liang Jiao.

**Writing – original draft:** Zhangjie Yang.

**Writing – review & editing:** Guojun Chen.

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
