## [Decision Letter · Decision Letter 0]

20 Feb 2026

PONE-D-25-63580A new target:  AlkBH2 promotes bladder cancer by upregulation of inflammationPLOS One

Dear Dr. Chen,

Thank you for submitting your manuscript to PLOS ONE. After careful consideration, we feel that it has merit but does not fully meet PLOS ONE’s publication criteria as it currently stands. Therefore, we invite you to submit a revised version of the manuscript that addresses the points raised during the review process.

If applicable, we recommend that you deposit your laboratory protocols in protocols.io to enhance the reproducibility of your results. Protocols.io assigns your protocol its own identifier (DOI) so that it can be cited independently in the future. For instructions see: https://journals.plos.org/plosone/s/submission-guidelines#loc-laboratory-protocols. Additionally, PLOS ONE offers an option for publishing peer-reviewed Lab Protocol articles, which describe protocols hosted on protocols.io. Read more information on sharing protocols at . Additionally, PLOS ONE offers an option for publishing peer-reviewed Lab Protocol articles, which describe protocols hosted on protocols.io. Read more information on sharing protocols at https://plos.org/protocols?utm_medium=editorial-email&utm_source=authorletters&utm_campaign=protocols..

We look forward to receiving your revised manuscript.

Kind regards,

Soumen Bera, PhD

Academic Editor

PLOS One

Journal Requirements:

https://journals.plos.org/plosone/s/file?id=. wjVg/PLOSOne_formatting_sample_main_body.pdf and=. wjVg/PLOSOne_formatting_sample_main_body.pdf and

https://journals.plos.org/plosone/s/file?id=ba62/PLOSOne_formatting_sample_title_authors_affiliations.pdf..

“Qinghai Key Construction Project of Specialized Departments of Qinghai University Affiliated Hospital (Qinghai Weijianwei [2023]133)”

3. Please be informed that funding information should not appear in the Acknowledgments section or other areas of your manuscript. We will only publish funding information present in the Funding Statement section of the online submission form. Please remove any funding-related text from the manuscript.

5. Please include captions for your Supporting Information files at the end of your manuscript, and update any in-text citations to match accordingly. Please see our Supporting Information guidelines for more information: http://journals.plos.org/plosone/s/supporting-information..

6. PLOS ONE now requires that authors provide the original uncropped and unadjusted images underlying all blot or gel results reported in a submission’s figures or Supporting Information files. This policy and the journal’s other requirements for blot/gel reporting and figure preparation are described in detail at https://journals.plos.org/plosone/s/figures#loc-blot-and-gel-reporting-requirements and https://journals.plos.org/plosone/s/figures#loc-preparing-figures-from-image-files. When you submit your revised manuscript, please ensure that your figures adhere fully to these guidelines and provide the original underlying images for all blot or gel data reported in your submission. See the following link for instructions on providing the original image data: s://journals.plos.org/plosone/s/figures#loc-original-images-for-blots-and-gels.

Additional Editor Comments (if provided):

Thank you for submitting your manuscript to Plos One. We have now received evaluations from two independent reviewers, and I have carefully considered their comments alongside my own assessment of the work.

After due deliberation, I have decided that the manuscript requires "major revision" before it can be considered further for publication. While one reviewer found the work promising and raised only minor concerns, the second reviewer has identified a number of substantive issues that must be satisfactorily addressed.

If you choose to revise and resubmit, please provide a detailed point-by-point response letter addressing each reviewer comment, with specific reference to the changes made in the manuscript.

The full reviewer reports are appended below for your reference.

We look forward to receiving your revised manuscript.

Reviewers' comments:

Reviewer's Responses to Questions

**Comments to the Author**

1. Is the manuscript technically sound, and do the data support the conclusions?

Reviewer #1: No

Reviewer #2: Yes

2. Has the statistical analysis been performed appropriately and rigorously? 

Reviewer #1: Yes

Reviewer #2: Yes

3. Have the authors made all data underlying the findings in their manuscript fully available?

Reviewer #1: No

Reviewer #2: Yes

4. Is the manuscript presented in an intelligible fashion and written in standard English?

Reviewer #1: No

Reviewer #2: Yes

5. Review Comments to the Author

Reviewer #1: The study addresses an interesting and potentially important biological question; however, substantial concerns regarding experimental rigor, methodological transparency, and manuscript organization currently limit the strength and interpretability of the conclusions. Major revisions are required before the work can be considered further.

Major Concerns

1. Experimental Design and Methodological Rigor

o The use of the CCK-8 assay as a surrogate for cell proliferation is a significant limitation, as this assay primarily reflects metabolic activity rather than true proliferative capacity. The inclusion of proliferation-specific assays (such as BrdU/EdU incorporation, Ki-67 staining, or real-time live-cell imaging platforms like IncuCyte) is strongly recommended. If these experiments cannot be added, the limitation should be explicitly acknowledged and discussed.

o Migration and wound-healing assays were performed without mitotic inhibitors (e.g., mitomycin C or colchicine). As a result, it is unclear whether the observed effects reflect changes in cell motility or are confounded by differences in cell proliferation. This issue critically undermines the interpretation of the migration data and should be addressed experimentally or discussed as a limitation.

o The claim of NF-κB pathway enrichment requires stronger supporting evidence. Pathway enrichment analysis results (e.g., KEGG pathway analysis) should be presented in the form of figures or heat maps to substantiate this conclusion.

o While NF-κB colocalization studies are presented, nuclear localization should be more rigorously validated using biochemical approaches. Western blot analysis with nuclear and cytoplasmic fractionation, including a nuclear marker such as lamin B1, would strengthen the evidence for NF-κB nuclear translocation.

o Many of the study’s conclusions rely on a single experimental approach per biological claim. Greater experimental rigor is needed by validating key findings using complementary or orthogonal methods.

o In Figure 2A, transfection efficiency is assessed by quantifying GFP-positive cells. However, GFP signals are also visible in the Control and AlkBH2− panels. This observation requires clarification unless genetically modified GFP-expressing T24 and TCC cell lines were used as controls. If that has been the case, then the reason behind this selection should also be discussed in the manuscript.

o The transfection experiments would benefit from the inclusion of appropriate controls, such as scrambled RNA controls. Additionally, the use of pharmacological inhibitors targeting AlkBH2 would provide important validation for the observed effects of AlkBH2 overexpression and help establish specificity.

o Quality of the figures needs to be improved, specifically Fig 3B, 4A, 4E are grainy and denies interpretation.

2. Insufficient Methodological Details

o Key reagent information is inconsistently reported. Manufacturer names, catalog numbers, and RRIDs for antibodies, cell lines, and critical reagents should be provided throughout to ensure reproducibility.

o The protocol used for generating conditioned media in angiogenesis assays is insufficiently described. Details such as cell density, duration of conditioning, serum conditions, and normalization strategies should be clearly specified.

3. Manuscript Organization and Flow

o Some of the subsection titles in the Materials and Methods section do not accurately reflect the content. For example, protocols related to cell cycle phase analysis are described under cell proliferation, which may confuse readers.

o The sequence of experiments described in the Materials and Methods does not align with the order of results presented. Reorganizing this section to mirror the Results would significantly improve readability.

o The Discussion section contains redundancy and would benefit from streamlining. A more focused discussion that directly links the study’s key findings to existing literature, while clearly highlighting novelty and limitations, is encouraged.

Minor Comments

• The manuscript requires careful language editing to improve clarity, grammar, and overall readability.

• All abbreviations (e.g., FA for formic acid) should be defined at first mention in the text.

• Formatting of superscripts, subscripts, and symbols is inconsistent and should be standardized throughout the manuscript.

Reviewer #2: DNA demethylase AlkB homolog 2 (AlkBH2) has not been well characterized in the context of inflammatory signaling or bladder cancer biology. In this regard, the authors propose a potentially interesting link between AlkBH2 and inflammation-driven bladder cancer progression. However, given the established complexity of inflammation-cancer crosstalk, careful and rigorous presentation of supporting evidence are essential to substantiate the claimed oncogenic and pro-inflammatory role of AlkBH2.

Major point:

(1) It is not clear whether demethylase activity is required for driving the inflammation. AlkBH2 has well-characterized HDH catalytic motif. Authors should mutate AlkBH2 and perform at least a few experiment of Figure 5.

(2) It is unclear how the conclusions drawn from the proteomic (mass spectrometry) analysis are supported, as no figure or data visualization corresponding to these results is presented in the manuscript. Given that mass spectrometry–based proteomics typically generates high-dimensional datasets requiring careful quality control, normalization, and statistical filtering, the absence of a dedicated figure (e.g., workflow schematic, volcano plot, heatmap, or pathway enrichment visualization) raises concerns about data transparency and reproducibility. At minimum, access to the raw or processed datasets should be provided. Without such evidence, it is difficult to independently assess the robustness of the proteomic analysis or the validity of the claim that AlkBH2 activates the NF-κB pathway.

Minor points:

1) Different font and different size fonts should be corrected

2) Fig 8, panel A, B: why there is no DAPI stained cells in panel A and B?

3) Details of the q-PCR primers and elisa sample preparation should be provided

4) Details of clones for silencing and overexpression should be provided. If they were obtained form elsewhere, sources should be mentioned.

5) Lane 350-350: this section should be covered in the introduction

6. PLOS authors have the option to publish the peer review history of their article (what does this mean?). If published, this will include your full peer review and any attached files.). If published, this will include your full peer review and any attached files.

.

Reviewer #1: No

Reviewer #2: **Yes:** Roy AnindyaRoy Anindya

To ensure your figures meet our technical requirements, please review our figure guidelines: s://journals.plos.org/plosone/s/figures

You may also use PLOS’s free figure tool, NAAS, to help you prepare publication quality figures: s://journals.plos.org/plosone/s/figures#loc-tools-for-figure-preparation.

---

## [Author Response · Author response to Decision Letter 1]

8 Mar 2026

Dear editor,

Thank you for your suggestions. We will carefully revise each suggestion and provide responses.

1. We have revised the manuscript in accordance with the formatting requirements of the PLOS ONE journal.

2. “The funders had no role in study design, data collection and analysis, decision to publish, or preparation of the manuscript.” We have added this Role of Funder statement in cover letter.

3. We have removed any funding-related text from the manuscript.

4. We have removed any ethics statement text besides the Methods from the manuscript.

5. According to Supporting Information guidelines, we have amended the caption.

6. We have uploaded original uncropped and unadjusted images underlying all blot or gel results reported in Supporting Information, and note this information in cover letter.

To reviewer1

Thank you for your all suggestions, which is extremely valuable and has greatly benefited us.

1. Thank you very much for your guidance. Due to the limitations of technology, your suggestions made us realize for the first time that there are so many methods for detecting cell proliferation. We have supplemented the real-time cell analysis experiment to replace CCK-8.

2. Thank you very much for your reminder. We did add mitomycin in the scratch test, but due to our writing oversight, it was not described in the methods section. We have already made the necessary revisions.

3. Thank you very much for your suggestion. As the results of proteomics sequencing were not satisfactory, we did not include them. Now, following your advice, we have added the volcano plot, heat map, and enrichment plot, among other results, to the figures.

4. Thank you very much for your suggestion. In order to ensure that the NF-κB nuclear localization mechanism is more strictly regulated, we have taken your advice into account and conducted additional experiments. We extracted nuclear proteins and used Lamin B1 as the marker to assess the relative nuclear entry of NF-κB.

5. Thank you for your comment regarding the transfection rate. This was our negligence in method as described. We have now added the method for calculating the transfection rate. First, count the cells using white light, then in the same field of view, calculate the GFP-positive cells using fluorescence, and the resulting ratio is the transfection rate.

6. Thank you for your comments regarding the inhibitor of AlkBH2. We fully agree with your suggestions. However, we are unable to supplement this experiment due to the time and economic costs involved. Although this may affect the completeness of the article, our experiment includes a group that knocked down AlkBH2, whose effect may be like its inhibitor. This should be able to contrast sharply with the effects observed from AlkBH2 overexpression, highlighting the role of AlkBH2 in bladder cancer.

7. Thank you for your suggestions regarding the quality of some of the images. In order to improve the image quality, we conducted new cloning experiments and Transwell experiments. However, for the scratch experiment, due to the limitations of the microscope, it was difficult to capture a very clear image using white light. Therefore, we used black-and-white images to highlight the shape of the scratch.

8. Thank you for your feedback. To ensure the reproducibility of the experiment, we have provided the information of the key reagents, including antibodies, cell lines, and the manufacturer names, product numbers of the key reagents, in the attachment.

9. Thank you very much for your suggestions. We have revised some of the materials and methods, especially regarding the analysis plan for the cell cycle and the detailed steps of the angiogenesis experiment, such as cell density, pre-treatment duration, serum conditions, and standardization strategies, etc., all of which have been clearly clarified.

10. Thank you very much for your suggestion. We have reorganized the materials and methods section to match the sequence of the "results" section.

11. Thank you very much for your suggestion. We have revised the discussion section and streamlined it. We focused on targeted discussions, directly connecting the key findings of the research with existing literature, while clearly highlighting its novelty and limitations.

12. Thank you very much for your feedback. We sought assistance from native English speakers and made revisions to the English grammar to make it more in line with the requirements for publication in the magazine.

13. Thank you very much for your suggestion. We have defined the first occurrence of the abbreviation.

14. Thank you very much for your feedback. We have carefully standardized the formats of superscripts, subscripts and symbols.

To reviewer2

Thank you for your suggestions, which is very helpful in improving the quality of this article.

1. Thank you very much for your suggestion regarding observing the effect of this gene on inflammation after mutation in AlkBH2. We totally agree with this method directly proved the relationship between AlkBH2 and validation. However, due to the limitations of funds and time, we only verified the above relationship from a side perspective. In Figure 5, overexpression of AlkBH2 clarified that AlkBH2 promotes inflammation in bladder cancer; while knocking down the AlkBH2 gene inhibited the inflammatory response, once again demonstrating the relationship between AlkBH2 and tumor inflammation. We hope that the "knockdown experiment" can, to certain extent, replace the AlkBH2 mutation experiment and make up for the shortcomings of this study. Of course, we also indicated our shortcomings in the article.

2. Thank you very much for your suggestion regarding the proteomics results. Due to the unsatisfactory outcome of the proteomics sequencing, we did not include it. Now, after listening to your suggestions, we have placed the volcano plot, heat map, and enrichment plot, among others, into the figures. This provides strong evidence to prove the validity of the statement that AlkBH2 activates the NF-κB pathway.

3. Thank you very much for your feedback. We carefully standardized the formatting for different sizes of fonts.

4. Thank you very much for your feedback. DAPI is present in the picture, but it needs to be enlarged for a clear view. Readers can enlarge the pictures for reading without being affected by this. The reason why it was not immediately found at first glance is as follows: 1) The third group is the AlkBH2 knockdown group. Under the same time conditions, cell growth was significantly inhibited, so it was very scarce. 2) To compare at the same level, the fluorescence shooting conditions cannot be changed, so the knockdown group with fewer cells had to be sacrificed, resulting in a relatively reduced absorption of blue light. In the future, we will overcome these limitations and improve our research techniques.

5. Thank you very much for your feedback. To ensure the reproducibility of the experiment, the information of key reagents has been provided, including antibodies, cell lines, and the manufacturer names, product numbers of the key reagents, as well as the relevant primer sequences used for PCR, which are included in the attachment.

6. Thank you very much for your comments. The silent and overexpressed lentiviruses were purchased from the company. We have indicated in the methods that they were obtained from GenePharma Company.

7. Thank you very much for your suggestion. We have incorporated the content you provided into the preface. To make the text more readable for the readers, we have revised the introduction and discussion sections, simplifying them and directly connecting the key findings of the research with the existing literature. At the same time, we have clearly highlighted its novelty and limitations.

At last, based on the opinions of the editors and reviewers, we carefully answered each question one by one, and made every effort to supplement additional experiments and provide explanations to address the relevant issues in order to improve the quality of the article. We hope to meet the requirements of the journal and be successfully accepted.

Best wishes,

Prof Chen

---

## [Decision Letter · Decision Letter 1]

6 Apr 2026

PONE-D-25-63580R1A new target:  AlkBH2 promotes bladder cancer by upregulation of inflammationPLOS One

Dear Dr. Chen,

Thank you for submitting your manuscript to PLOS ONE. After careful consideration, we feel that it has merit but does not fully meet PLOS ONE’s publication criteria as it currently stands. Therefore, we invite you to submit a revised version of the manuscript that addresses the points raised during the review process.

If applicable, we recommend that you deposit your laboratory protocols in protocols.io to enhance the reproducibility of your results. Protocols.io assigns your protocol its own identifier (DOI) so that it can be cited independently in the future. For instructions see: https://journals.plos.org/plosone/s/submission-guidelines#loc-laboratory-protocols. Additionally, PLOS ONE offers an option for publishing peer-reviewed Lab Protocol articles, which describe protocols hosted on protocols.io. Read more information on sharing protocols at . Additionally, PLOS ONE offers an option for publishing peer-reviewed Lab Protocol articles, which describe protocols hosted on protocols.io. Read more information on sharing protocols at https://plos.org/protocols?utm_medium=editorial-email&utm_source=authorletters&utm_campaign=protocols..

As the corresponding author, your ORCID iD is verified in the submission system and will appear in the published article. PLOS supports the use of ORCID, and we encourage all coauthors to register for an ORCID iD and use it as well. Please encourage your coauthors to verify their ORCID iD within the submission system before final acceptance, as unverified ORCID iDs will not appear in the published article. *Only* the individual author can complete the verification step; PLOS staff the individual author can complete the verification step; PLOS staff *cannot* verify ORCID iDs on behalf of authors.verify ORCID iDs on behalf of authors.

We look forward to receiving your revised manuscript.

Kind regards,

Soumen Bera, PhD

Academic Editor

PLOS One

Journal Requirements:

**Additional Editor Comments:**

Thank you for submitting your revised manuscript and response to the reviewers’ comments. We appreciate the time and effort invested in revising the work.

While one reviewer considers their comments to have been satisfactorily addressed, another reviewer and the editorial assessment find that important issues remain unresolved in the current revision. Although progress has been made, certain aspects of the manuscript would benefit from greater clarity, consistency, and completeness. In addition, some points raised in the reviewers’ feedback do not yet appear to be fully reflected in the revised manuscript, despite being noted as addressed in the response letter.

We therefore invite you to further revise the manuscript, ensuring that:

All remaining reviewer comments are addressed in a clear and comprehensive manner;

The response letter accurately and transparently reflects the changes made in the manuscript;

Any remaining ambiguities in methods, results, or interpretation are resolved to improve overall rigor and transparency.

If specific reviewer comments cannot be fully addressed due to methodological constraints, or other justified reasons, these should be clearly acknowledged and explained, either directly in the manuscript (e.g., in the Discussion or Methods) and/or explicitly in the rebuttal letter. Providing a clear rationale in such cases will help facilitate the editorial and peer review process.

Please provide a detailed, point‑by‑point response outlining how each comment has been addressed or justified in the revised submission.

Reviewers' comments:

Reviewer's Responses to Questions

**Comments to the Author**

1. If the authors have adequately addressed your comments raised in a previous round of review and you feel that this manuscript is now acceptable for publication, you may indicate that here to bypass the “Comments to the Author” section, enter your conflict of interest statement in the “Confidential to Editor” section, and submit your "Accept" recommendation.

Reviewer #1: (No Response)

Reviewer #2: All comments have been addressed

2. Is the manuscript technically sound, and do the data support the conclusions?

Reviewer #1: Partly

Reviewer #2: Yes

3. Has the statistical analysis been performed appropriately and rigorously? 

Reviewer #1: N/A

Reviewer #2: Yes

4. Have the authors made all data underlying the findings in their manuscript fully available?

Reviewer #1: Yes

Reviewer #2: Yes

5. Is the manuscript presented in an intelligible fashion and written in standard English?

Reviewer #1: Yes

Reviewer #2: Yes

6. Review Comments to the Author

Reviewer #1: The authors have included real-time cell proliferation assay and proteomic analysis like volcano plots which have considerably strengthened the manuscript. However, several critical concerns remain insufficiently addressed. In particular, issues related to methodological transparency, experimental rigor, and consistency in reporting limit the reliability and reproducibility of the findings.

1. The authors are advised to carefully reconsider and revise the methodology of the wound-healing assay. As noted in the earlier review, performing this assay in the presence of mitosis inhibitors (e.g., mitomycin C) is important to distinguish cell migration from proliferation effects. If the authors choose to conduct the assay without mitomycin C, they must provide a clear scientific justification and explicitly discuss the associated limitations.

2. In the endothelial cell tube formation assay, the authors refer to the use of “normalized CM” without explaining how the conditioned media (CM) was generated. It is assumed that CM refers to media conditioned by a specific cell line; however, no methodological details have been provided, despite this issue being raised previously. The authors should include a clear and detailed description of CM preparation and normalization procedures.

3. Authors are requested, again, to explain the source of the fluorescence observed in the Control and AlkBH2- panels of Fig 2A.

4. The authors should present the data presented in Fig 3A as quantitative measurements at defined time points, accompanied by appropriate statistical parameters (e.g., mean ± SD). This would provide a more rigorous, interpretable, and scientifically sound representation of the results.

5. A consistent approach to data representation should be maintained. In Figure 5, the authors have used both horizontal and vertical bar graphs to present similar ELISA data. This inconsistency may affect readability and interpretation. The authors should either standardize the graph format or provide a clear scientific rationale for using different orientations.

6. The authors are strongly advised to maintain a consistent style throughout the manuscript. It is standard practice to provide complete manufacturing details for all reagents used, including their RRIDs, to ensure reproducibility. While the authors acknowledged this requirement in their response, the corresponding revisions have been implemented only sporadically. Although Table S1 includes details for ELISA kits, antibodies, and primer sequences, important information is still missing for reagents used in cell culture, staining, flow cytometry, wound healing assays, and colony formation assays. The authors should comprehensively update this information across all experimental sections.

Reviewer #2: All the questions answered satisfactorily. Manuscript and figures has been revised sufficiently. Some suggested experiments could not be done due constrains.

7. PLOS authors have the option to publish the peer review history of their article (what does this mean?). If published, this will include your full peer review and any attached files.). If published, this will include your full peer review and any attached files.

.

Reviewer #1: No

Reviewer #2: **Yes:** Anindya RoyAnindya Roy

To ensure your figures meet our technical requirements, please review our figure guidelines: s://journals.plos.org/plosone/s/figures

You may also use PLOS’s free figure tool, NAAS, to help you prepare publication quality figures: s://journals.plos.org/plosone/s/figures#loc-tools-for-figure-preparation.

---

## [Author Response · Author response to Decision Letter 2]

7 Apr 2026

Dear editor,

Thank you very much for your recognition of our work. We have made detailed revisions according to your suggestions to meet the requirements of your journal.

1. According to your advice, we have uploaded original data to Protocols.io. (DOI): https://www.protocols.io/file/be2wqczccx.zip

2. We are verifying our ORCID iD within the submission system before final publish.

To reviewer1

Thank you for all the comments you have made on our article. Your opinions not only corrected our mistakes but also significantly improved the quality of the article.

1. I'm very sorry for my oversight that caused you to misunderstand. In the last response, we clearly stated that we had added mitomycin C. However, because English is not our native language, we used "without" in the article revision, while our original intention was "with". Well, I admit that I am a rather careless person. Once again, thank you for your carefulness and rigor in pointing out our mistakes.

2. Thank you for your feedback. Standardized CM refers to the preparation of a standard mixed culture medium containing factors secreted by bladder cancer cells, which is used to stimulate HUVEC and evaluate its angiogenic promoting ability. Here, "standardized" means that we use a unified cell count of 106 and uniformly culture for 48 hours to obtain the mixed culture medium containing the factors secreted by bladder cancer cells. The corresponding explanation has been supplemented in the methods section.

3. Thank you very much for your question. All our groups, including the control group and the ALKBH2+ and ALKBH2- groups, were transfected with lentivirus carrying the green fluorescent protein gene (GFP). The difference is that the control group was only transfected with the GFP gene, while the ALKBH2+ and ALKBH2- groups were transfected with the GFP gene in addition to overexpressed and interfered genes. Therefore, all three groups of cells have green fluorescence that were observed in Fig 2A, in order to ensure the consistency of the experiment. The corresponding explanations have been supplemented in the methods section.

4. Thank you very much for your suggestion. In the real-time cell proliferation experiment, a mean ± SD value is obtained at extremely short time intervals. In Fig 3A, we plotted the graph based on each mean ± SD value. By zooming in, we can observe that it is composed of countless independent mean ± SD points and lines. The original data including mean ± SD value has been uploaded to the editing department. After the article is officially published, readers should be able to access it through the link.

5. Thank you very much for your suggestion. The reason why the bar graphs in Fig 5 are not uniform is due to the layout issue. If all were vertical bars, it would not be possible to place the ELISA results in one figure. Considering the logic and continuity of the article, we did our best to place the ELISA results in one figure, which led to some bar charts having to be horizontal bars. At the same time, we ingeniously distinguished the types of inflammatory factors through different types. All vertical bars represent pro-inflammatory factors, while the horizontal bars represent factors that inhibit inflammation. This not only solves the layout issue but also enhances the readers' interest in reading to a certain extent.

6. Thank you very much for your feedback. We originally thought that listing only the main reagents would suffice. We have now provided detailed information about all the reagents in Table S1.

To reviewer2

Thank you very much for your appreciation of this article. Your recognition and encouragement are the greatest motivation for our progress. Thank you again!

Best wishes,

Prof Guojun Chen

---

## [Editor Report · Decision Letter 2]

13 Apr 2026

A new target:  AlkBH2 promotes bladder cancer by upregulation of inflammation

PONE-D-25-63580R2

Dear Dr. Chen,

We’re pleased to inform you that your manuscript has been judged scientifically suitable for publication and will be formally accepted for publication once it meets all outstanding technical requirements.

An invoice will be generated when your article is formally accepted. Please note, if your institution has a publishing partnership with PLOS and your article meets the relevant criteria, all or part of your publication costs will be covered. Please make sure your user information is up-to-date by logging into Editorial Manager at Editorial Manager® and clicking the ‘Update My Information' link at the top of the page. For questions related to billing, please contact  and clicking the ‘Update My Information' link at the top of the page. For questions related to billing, please contact billing support..

Kind regards,

Soumen Bera, PhD

Academic Editor

PLOS One
---

## [Editor Report · Acceptance letter]

PONE-D-25-63580R2

PLOS One

Dear Dr. Chen,

I'm pleased to inform you that your manuscript has been deemed suitable for publication in PLOS One. Congratulations! Your manuscript is now being handed over to our production team.

Kind regards,

on behalf of

Dr. Soumen Bera

Academic Editor

PLOS One